# Space-Time Dependence of Emotions on Twitter after a Natural Disaster

**DOI:** 10.3390/ijerph18105292

**Published:** 2021-05-16

**Authors:** Sonja I. Garske, Suzanne Elayan, Martin Sykora, Tamar Edry, Linus B. Grabenhenrich, Sandro Galea, Sarah R. Lowe, Oliver Gruebner

**Affiliations:** 1State Office of Health and Social Affairs, 10639 Berlin, Germany; sonja.garske@gmx.net; 2Centre for Information Management, Loughborough University, Leicestershire LE11 3TU, UK; S.Elayan2@lboro.ac.uk (S.E.); M.D.Sykora@lboro.ac.uk (M.S.); 3Department of Geography, University of Zurich, 8057 Zurich, Switzerland; tamar.edry@uzh.ch; 4Department for Methodology and Research Infrastructure, Robert Koch-Institut, 13359 Berlin, Germany; linus.grabenhenrich@charite.de; 5Department of Dermatology, Venerology and Allergology, Charité—Universitätsmedizin Berlin, 10117 Berlin, Germany; 6School of Public Health, Boston University, Boston, MA 02118, USA; sgalea@bu.edu; 7Department of Social & Behavioral Sciences, Yale School of Public Health, New Haven, CT 06510, USA; sarah.lowe@yale.edu; 8Epidemiology, Biostatistics and Prevention Institute, University of Zurich, 8001 Zurich, Switzerland

**Keywords:** natural disaster, post disaster mental health, digital epidemiology, spatial epidemiology, health geography, Twitter

## Abstract

Natural disasters can have significant consequences for population mental health. Using a digital spatial epidemiologic approach, this study documents emotional changes over space and time in the context of a large-scale disaster. Our aims were to (a) explore the spatial distribution of negative emotional expressions of Twitter users before, during, and after Superstorm Sandy in New York City (NYC) in 2012 and (b) examine potential correlations between socioeconomic status and infrastructural damage with negative emotional expressions across NYC census tracts over time. A total of 984,311 geo-referenced tweets with negative basic emotions (anger, disgust, fear, sadness, shame) were collected and assigned to the census tracts within NYC boroughs between 8 October and 18 November 2012. Global and local univariate and bivariate Moran’s I statistics were used to analyze the data. We found local spatial clusters of all negative emotions over all disaster periods. Socioeconomic status and infrastructural damage were predominantly correlated with disgust, fear, and shame post-disaster. We identified spatial clusters of emotional reactions during and in the aftermath of a large-scale disaster that could help provide guidance about where immediate and long-term relief measures are needed the most, if transferred to similar events and on comparable data worldwide.

## 1. Introduction

Large-scale natural disasters, such as exceptionally severe storms, will continue to become more frequent against the backdrop of increasingly apparent climate change. Large-scale natural disasters have substantial mental health consequences [1,2]. The most common documented psychological problems in this context are depression and posttraumatic stress disorder (PTSD) [1,2,3,4,5,6,7]. Mental health consequences vary across socio-demographic groups and neighborhoods depending on local social factors [8,9]. For example, in the aftermath of Superstorm Sandy, New York City (NYC), demographic factors (older age, non-Hispanic black) and more disaster-related stressors were significantly associated with higher posttraumatic stress [10]. Further, living in communities with higher economic development was associated with lower risk of depression [8]. However, classical disaster epidemiological studies are usually conducted only after the disaster, and depend on self-report data that are subject to several biases, particularly recall bias [1,9].

One way to overcome potential recall bias is to use social media data, which can provide a near real-time record of emotions after a disaster. Geo-referenced Twitter data that are freely and quickly accessible can provide an inexpensive supplement to conventional epidemiological survey methods [11,12,13,14,15,16]. In 2010, it was shown that, during and immediately after a natural disaster, Twitter users shared posts narrating their experiences [17]. Further, Twitter provides a time stamp for every single tweet. This time aspect could be valuable for both the provision of emergency management [18,19] and disaster epidemiological research, as it also allows for the investigation of periods before and during a disaster in addition to its aftermath [14,16,20,21,22]. In addition, studies have revealed that linguistic features of trauma reports are correlated with the development of depression and PTSD [23] and that these may first appear in emotional expressions [24].

With the help of advanced sentiment analysis software tools, the emotional expressions within the recorded tweets can be evaluated and thereby provide valuable insights in mental health research [14,16,20,21,22]. For example, tweets may provide information about the extent and strength of basic emotions, such as anger, disgust, fear, happiness, sadness, and surprise, as defined by Ekman [25,26], and crucial knowledge about how traumatic events are experienced during the event itself and its immediate and long-term aftermath [14]. In order to investigate such emotional expressions in the context of disaster, scholars have already used geo-referenced Twitter data to find out how negative emotional expressions are distributed across space and time [14,16,24]. For example, Gruebner and colleagues found that negative emotional reactions as identified in Twitter were concentrated in specific NYC neighborhoods during and after Superstorm Sandy, particularly on Staten Island [14]. In addition, they found that negative emotions in census tracts were correlated over time, with high proportions of negative emotions before the storm being positively associated with negative emotions after the storm. Another study by Gruebner and colleagues found that during Superstorm Sandy, specific negative emotions clustered over single days, highlighting perceived locally specific health risks, such as falling trees or rattling windows [24]. While other studies have also applied geo-referenced social media data in the context of disaster events [14,15,16,17,18,20,21,27,28,29,30,31,32,33,34], none to our knowledge has investigated the socio-demographic context in which emotions have been expressed. Further, as far as we know, the relationship between infrastructural damage and negative emotional expressions on Twitter before, during, or after a natural disaster has not been investigated. Therefore, we do not know much about whether socio-demographic factors or infrastructural damage have an impact on the spatial distribution of negative emotions as seen on Twitter during a natural disaster.

We used NYC Twitter data in the context of Superstorm Sandy that hit the New York City area 29 October 2012 to (1) explore the spatial distribution of multiple negative emotions of Twitter users in NYC census tracts before, during, and after Superstorm Sandy and to (2) examine spatial associations of socio-economic status, infrastructural damage, and negative emotional expressions in Twitter tweets across NYC census tracts over time.

## 2. Materials and Methods

### 2.1. Data and Study Population

The study draws on publicly available English-language Twitter tweets that were posted in New York City (NYC) between 8 October 2012 and 18 November 2012, detailed elsewhere [14]. Briefly, the tweets were retrieved from the ‘Geo-Tweet’ archive of the ‘Harvard Centre for Geographical Analysis’ (CGA) and missing data were supplemented by data from ‘Geofeedia’ [14]. Previously, an emotion analysis of the data was carried out with EMOTIVE (Extracting the Meaning Of Terse Information in a Visualization of Emotion), a software that detects and measures emotions in social media posts [35]. This software was used to extract six basic emotions defined by Ekman (anger, disgust, fear, happiness, sadness and surprise) [25,26] as well as shame and confusion [35]. Subsequently, the data were cleaned from social bots using a threshold of above 0.5 for the Botometer-estimated probability score (cap universal), to exclude tweets produced by automated accounts [36]. A total of *N* = 3852 such accounts were excluded from the dataset. Since only geo-referenced tweets were studied, VPN users can generally be ruled out, while proxies, such as used for bots and other inauthentic user account automation (i.e., to avoid being blocked by Twitter) would have been removed in the above Botometer filtering step.

The current study only used tweets with negative emotional expressions: anger (*N* = 4768, 0.48%), disgust (*N* = 6927, 0.70%), fear (*N* = 3738, 0.38%), sadness (*N* = 13,963, 1.42%), as well as shame (*N* = 1046, 0.11%), which were additionally combined to one variable of all these negative expressions NeEm (*N* = 30,442, 3.09%). The data were then aggregated at *N* = 2165 NYC census tracts [37] and fully anonymized, since all personal identifiers were deleted during the aggregation process. Specifically, each emotion was counted and a proportion calculated (i.e., number of specific emotions divided by all tweets) for each census tract and disaster period (pre-, peri-, and post-disaster). We defined three disaster periods with approximately 14 days each, to be consistent with the time span usually considered in psychiatric questionnaires [24,38,39]. In total, the dataset comprises *N* = 984,311 tweets of which *N* = 384,753 tweets were collected in the pre-disaster period (8 to 21 October), *N* = 228,293 in the peri-disaster period (22 October to 4 November), and *N* = 371,265 in the post-disaster period (15 to 18 November).

Three exposure factors were considered at the NYC census tract level [37] to express socio-economic status (SES) and infrastructural damage. For SES, we considered the unemployment rate in a CT, which was available from the American Community Survey (ACS_10_5YR_DP03). The level of inundation and level of building destruction represented infrastructural damage in our analyses and were available from the Federal Emergency Management Agency (FEMA). According to FEMA, level of inundation had three categories. For our analysis, we generated three dummy variables for each of these inundation categories separately, coded as 1 indicating whether more than 50% (*N* = 317 CTs, 14.6% of NYC, zone 1), up to 50% (*N* = 309 CTs, 14.3% of NYC, zone 2), or none of a CT was flooded (*N* = 1540 CTs, 71.1% of NYC, zone 0) and 0 otherwise.

The degree of building destruction had four categories based on FEMA’s Building damage classification for each building structure, i.e., affected (general superficial damage), minor (solid structures sustained exterior damage), major (extensive structural damage), and destroyed (the structure has been completely destroyed). Again, we created dummy variables for each of these categories and aggregated the values representing the proportion of each category at a census tract.

### 2.2. Analysis

First, we tested for spatial dependence in the outcome variables and used the univariate global and local Moran’s Index separately over all time periods. The outcome variables included anger, disgust, fear, sadness, shame and NeEm. The Moran’s Index usually ranges between −1 and 1, with negative values indicating spatial dispersion and positive values indicating spatial clustering, while values close to 0 indicate spatial randomness. For the analysis, a spatial weights file was defined to identify which census tract is a neighbor of another census tract, using the queen contiguity for weighting.

The Local Indicator of Spatial Association (LISA) was used to identify local spatial autocorrelations in our data, in which each individual census tract is analyzed in the context of neighboring census tracts. In the course of this, it is determined whether the local pattern differs from the global pattern. The sum of the LISA for all observations is proportional to Moran’s I statistics. The LISA statistics has two different functions: Local cluster analysis and testing for outliers. We used it exclusively for local cluster analysis [40].

Second, we tested for local spatial correlations using the bivariate global and local Moran`s Index (I). Analogous to the univariate case, the bivariate LISA identifies clusters of above (or below) average spatial correlation. The bivariate local Moran’s I expresses the spatial lag between the outcome variables (anger, disgust, fear, sadness, shame and NeEm) in a census tract and the mean of the exposure variables in the neighboring census tract [40]. Specifically, we tested for correlations of the exposure variables unemployment rate, levels of building destruction (affected, minor, major, destroyed), and level of inundation (≤50%, >50%, 0% flooding) with the outcome variables, whereby level of inundation and level of building destruction were considered in the peri- and post-disaster periods only. All analysis was carried out using OpenGeoDa (Version 1.12.1.161) [41,42].

Good epidemiological practice includes careful compliance with data protection and ethical standards [43]. All Twitter data that we used for the analysis were completely anonymized and aggregated at CT level. The merging of Twitter data with socio-economic data was done using the Census Tract ID (BoroCT2010) as the connection key to link the different databases. A retracement to the individual is not possible under any circumstances. The use of these freely and fast accessible social media data, especially those that are anonymized and aggregated, legally complies with data protection and ethical regulations in the European Union where the data has been analyzed for this study.

## 3. Results

Generally, the number of tweets in the pre- and post-disaster period was approximately the same. In the peri-disaster period however, significantly fewer tweets were posted than in the pre-, and post-disaster periods. The number of tweets that included emotions sadness and fear increased slightly in the peri-disaster period. The emotion sadness was with 46% the most prevalent emotion in terms of absolute numbers of all negative emotions, with about 3%, shame being the least frequently expressed emotion.

### 3.1. Spatial Distribution of Emotions

We found very weak but statistically significant global spatial clustering in four emotions, i.e., anger (Moran’s I 0.025, *p*-value < 0.05), fear (0.0210, <0.05), sadness (0.0232, <0.05), and the combined set of negative emotions NeEm (0.0554, <0.001) in the pre-disaster period. For the post-disaster period, global clustering could be found for three emotions, disgust (0.0227, <0.05), shame (0.0202, <0.05), and NeEm (0.0855, <0.001) (Table 1).

For the pre-disaster period, the LISA maps in Figure 1 show that local spatial high-high clusters (above average values in a CT surrounded by CT’s with also above average values) of these emotions mainly concentrated in Brooklyn and Queens while individual clusters could also be found for the Bronx and Staten Island.

For the post-disaster period, the emotion disgust formed spatial clusters in Queens and in Brooklyn (Figure 2). CT 640 (Georgetown-Marine Park-Bergen Beach-Mill Basin) in Brooklyn is particularly interesting as it is the only CT where almost all emotions (fear, anger and disgust) spatially concentrate. We also found high clusters for the emotion shame in Queens and Brooklyn, and a single one in Manhattan (CT 238.01, Roosevelt Island). Furthermore, NeEm clustered primarily in Queens and Brooklyn with single clusters in Staten Island (CT 132.03, 156.03 Great Kills) and the Bronx (CT 123, Crotona Park East). In addition, we found that high-high clusters were stable over time. For example, the combined negative emotions NeEm clustered in Queens (273, Jackson Heights) from pre-disaster to post-disaster.

### 3.2. Spatial Associations of Exposure Factors and Emotions

We found global negative spatial associations for unemployment rate and NeEm in both pre-disaster (Moran’s I −0.0229, *p*-value <0.01) and post-disaster periods (−0.0228, <0.01), indicating a tendency towards spatial dispersion (see Table 2). While no statistically significant spatial associations could be found between inundation levels and NeEm in neither disaster period, we found that higher levels of affected buildings in a neighboring CT were positively and spatially associated with NeEm post-disaster (0.0432, <0.001). Higher levels of major building destruction were also positively associated with NeEm, but only post-disaster (0.0355, <0.01). In relation to individual emotions, very few significant spatial associations were found with the exposure variables. Predominantly in the post-disaster period, the emotions fear and disgust stand out here. Considering all variable combinations, Brooklyn was the Borough with the most high-high clusters detected.

Despite the dispersion patterns found in the global statistics, the local Moran’s I statistic identified positive spatial associations between above average unemployment and above average NeEm as indicated by high-high-clusters in Brooklyn and Queens pre- and post-disaster (Figure 3A,B). In Brooklyn, four clusters were constant over all disaster periods in areas with high unemployment rate. Three further clusters were constant over the peri- and post-disaster period in Brooklyn, Rugby-Remsen Village CT 938 (unemployment rate: 12.8%), CT 848 (unemployment rate: 17.5%) and Queens: CT 273 Jackson Heights (unemployment rate: 9.2%).

The level of building destruction also showed local spatial associations with NeEm and exhibited local clusters. Most clusters were found for the category affected followed by major. Two clusters showed persistence over all disaster periods and exhibited that affected buildings were locally and spatially associated with NeEm in Brooklyn, CT 268, Bensonhurst West (unemployment rate: 11.3%), and CT 258 Bensonhurst West. In the peri- and post-disaster period, five clusters were constant in Brooklyn, (CT 848, 938, 1010, 426 and 404) and one in Queens (CT 273).

A Appendix A shows results of the univariate Moran’s I for the exposure variables SES and infrastructural damage (global and local) for comparison purposes (Appendix A, not reported here).

## 4. Discussion

We found that negative emotions of Twitter users were spatially distributed across NYC census tracts with local spatial clusters in all emotions before, during, and after Superstorm Sandy. Further, this study is among the first to provide evidence for significant spatial associations of socio-economic status and infrastructural damage with negative emotional expressions in tweets exhibiting persistence over time.

### 4.1. Clusters of Negative Emotions

In the context of natural disasters, several studies have shown that geographically concentrated negative emotions could be identified before, during, and after a disaster with the help of Twitter data [14,24,34,44]. In our study, we focused on negative emotions and examined them individually and as a variable of all negative emotions in all time periods. Whereas Gruebner et al. showed a similar finding [14,24], the current study identified spatial clusters of individual emotions over time, which gives us a more complete picture of the emotional distribution in the population of NYC over space and time. Since the individual negative emotions investigated are indicators of how people assess and deal with severe natural events [14], it is important to know the different spatial distribution of individual emotions so that geographically targeted preventive measures tailored toward different emotional responses can be implemented quickly, efficiently, and effectively. It is of note that the spatial clusters of high values of the individual emotions overlapped only sporadically and exclusively in the time after the disaster. In Brooklyn (Park Cemetery), for example, we identified a large cluster in which an overlapping of the emotions sadness and disgust clearly emerged. It is notable that this cluster was observed in a census tract with a large cemetery, where people gather to attend a funeral. As such emotions are typically found in cemeteries, the results indicate that this method allows us to spatially determine quite locally specific emotions.

In the context of disaster relief, it is precisely such overlapping patterns that give an indication of where negative emotions accumulate and where targeted measures to ameliorate psychological distress might be most helpful. In addition to the above-mentioned studies, our study was able to highlight this overlap more clearly by examining the individual emotions separately.

### 4.2. Correlations of Exposure Factors and Negative Emotions in Space

Previous studies have investigated protective and exposure factors with regard to mental health in the context of disasters [8,9,10,29,45]. Some of these studies have been able to offer precise insights into which exposure factors caused mental stress after Superstorm Sandy and how they were spatially spread [9,45]. Against this background, we included socio-economic exposure factors in our study and were able to gain insights into how negative emotions were distributed in correlation with the socio-economic exposure factors described above, in particular in the peri- and post-disaster period in NYC, and which risk areas were found in each period. To the best of our knowledge, this is the first study to detect spatial correlations between socio-economic status (unemployment rate), infrastructure damage (level of building destruction and level of inundation), and negative emotions as seen in Twitter in the Superstorm Sandy context. As in the study by Lowe et al. [45], our used method allowed us to identify spatial clusters in southern Queens and Brooklyn and on Staten Island. In contrast to the above-mentioned study that adopted a retrospective epidemiological study approach, our method allowed us to identify risk areas also for the peri-disaster period for comparisons with the peri and post-disaster periods. This approach could be useful for rapid disaster relief and, if the relevant data is available, could possibly be applied to other disaster events in other settings.

Particularly in the post-disaster period, unemployment can be a risk factor for mental health disorders and this was the case after Superstorm Sandy [9,10,45]. We identified spatial clusters in regions with medium to high unemployment mainly in Brooklyn and Queens. According to Lowe and colleagues [45], residents living in these regions were at higher risk for PTSD as compared to other regions two years after Superstorm Sandy. Our results provide a quick indication of the specific census tracts in which high unemployment rates were associated with high negative emotions, without the need for a time- and cost intensive survey. We were able to identify four clusters for the correlation between unemployment and NeEm that were spatially stable in all disaster periods. These clusters were characterized by medium unemployment rates (around 9.4%), had little building destruction and were all located in distinct flood zones (from >50% flooded to no flooding). If such overlapping patterns are considered by policymakers, the population in these and similar risk areas could be well supported from the beginning to mitigate the potential psychological burden of the disaster.

### 4.3. Limitations

Our study also had a few limitations that can be addressed by future studies. First, the data are from 2012 and are therefore not recent. Mental health issues, such as PTSD and depression, can linger for long periods of times, even years, if not treated. Therefore, although the data is old in academic terms, the implications can still be relevant to the real live people who have experienced superstorm Sandy. This is especially true for population with low socio-economic status with fewer resources [1]. Twitter as a data source allows for the analysis of the long-term mental health consequences of superstorm sandy in affected users. Furthermore, Twitter is a fairly “prototypical” social platform in a lot of its features and affordances [46], which makes the application of the methodologies used in this study likely transfer well to similar in-the-moment geo-referenced text-based datasets. Future studies could therefore take into consideration changes in Twitter usage since our study to replicate our efforts using more recent data and across more heterogeneous areas/events around the world. Second, only tweets in English language were used. This excludes tweets in other languages that were posted in NYC during the period of the study. For broader data coverage, other languages commonly spoken in NYC areas, such as Spanish and Chinese, should be considered as well [47]. Third, technical challenges could have led to distortions in data collection of the used Twitter data: For example, it cannot be ruled out that the emotion recognition software EMOTIVE was not capable of correctly recognizing linguistic subtleties, such as sarcasm and irony [48]. Further, only 6.5% of tweets had emotions (detected by EMOTIVE), and even fewer (47.1% of all tweets with emotions) had negative emotions. In addition, our approach could only detect specific emotions or overall negative emotions and it will be interesting to also use software that can detect symptoms of e.g., depression [49] or direct expressions of acute stress in social media users over space and time [50]. Fourth, Twitter users are not representative of the US population in terms of age, gender, or education, and Twitter users who use the location service are not representative of the general Twitter population according to a study by Sloan et al. [51]. Another bias can be with frequent users, who receive more weight in the analysis. One can tweet the same emotion many times, where others may tweet only once. An optional answer to that can be that the intensity of tweeting can reflect the intensity of the emotion, but this may have to do with many other personality factors as well. All of this introduces an additional selection bias into the study that could potentially be addressed by studies employing a nested approach, that is, a multilevel approach with tweets nested within individuals. However, a wide range of demographic groups is represented on Twitter [51], so it is still a population worth examining in the context of this study. Although the geo-location service was set to default during 2012 and users needed to actively opt out if they wished their tweets not being geo-referenced, we expect that a vast amount of tweets were not geo-referenced [51] and hence were excluded in our study. Fifth, it should be noted that in some census tracts low numbers of tweets have been identified that could have led to unreliable results for some regions. Finally, while we could not yet control for further variables in the current study, it would be interesting to examine the influence of other factors, such as age, sex, and race/ethnicity in the Twitter population, since there is evidence that belonging to a social minority can be a risk factor for suffering mental stress disorders after a disaster [1,8,9].

## 5. Conclusions

In this study, we demonstrated how exploratory spatial data analysis can be used to combine geo-referenced Twitter data with publicly available datasets from official sources to identify spatial clusters of negative emotional expressions before, during, and after a natural disaster. We found that low socioeconomic status and infrastructural damage were predominantly correlated with local clusters of negative emotions post-disaster. It is thus possible to identify areas of increased need for help. Our approach is transferable to other regions and potentially also to different data sources given that there is an indication of time and space available. For example, spatio-temporal triangulation of social media streams with local information on the social and physical contextual situation may provide useful information to estimate the contagion of infectious diseases [52] or mental health consequences of natural and human made disasters [53]. Against the background of increasing use of digitally connected devices and available big social media data, such approaches may help design locally specific mental health relief measures especially during times of crisis and in disaster affected areas. Against the background of increasing use of digitally connected devices and available big social media data, such approaches may help design locally specific mental health relief measures especially during times of crisis and in disaster affected areas.

## Figures and Tables

**Figure 1 ijerph-18-05292-f001:**
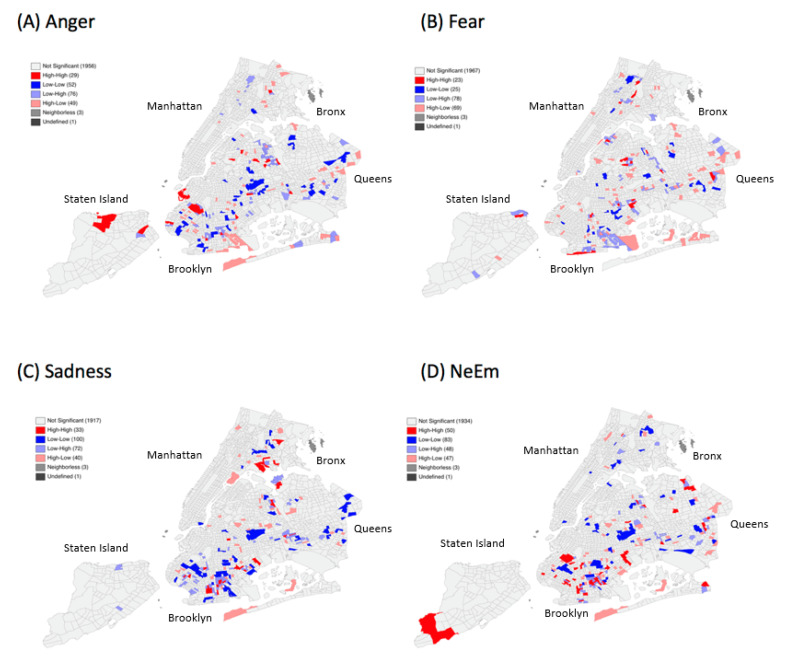
Univariate spatial clusters of (**A**) Anger, (**B**) Fear, (**C**) Sadness, and (**D**) the combined set of negative emotions (NeEm) pre-disaster.

**Figure 2 ijerph-18-05292-f002:**
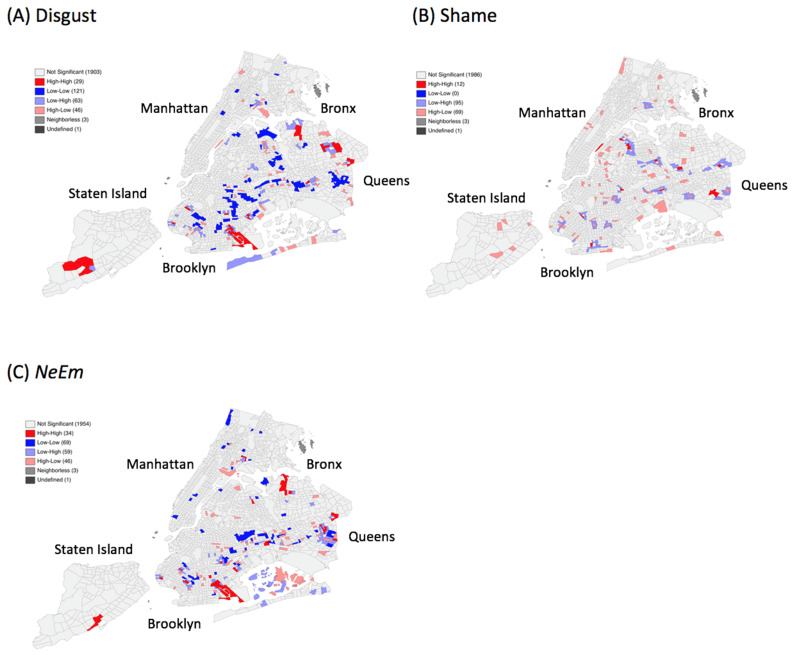
Univariate spatial clusters of (**A**) Disgust, (**B**) Shame, and (**C**) the combined set of negative emotions (*NeEm*) post-disaster.

**Figure 3 ijerph-18-05292-f003:**
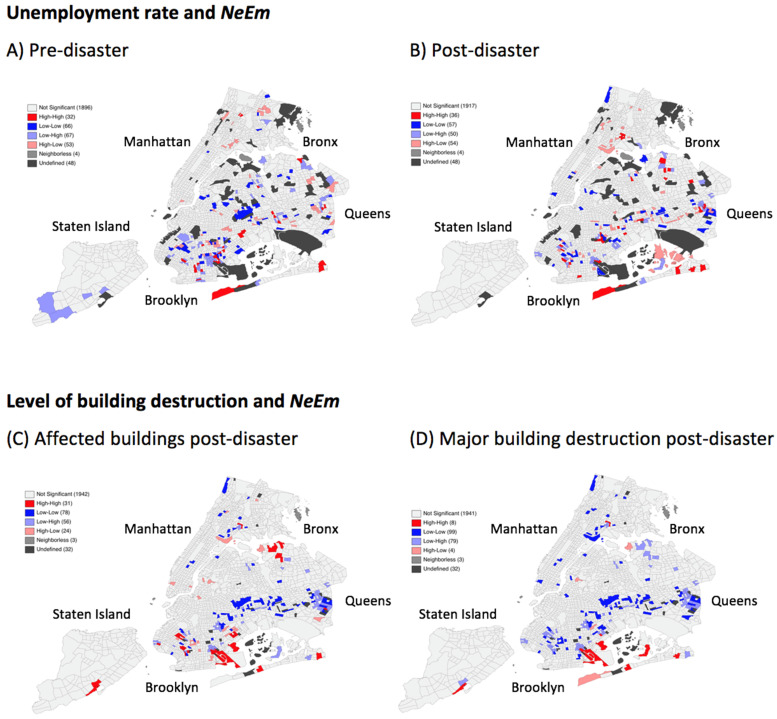
Bivariate spatial clusters exhibiting local spatial associations between exposure factors unemployment rate (**A**,**B**) and levels of building destruction (**C**,**D**) with the combined set of negative emotions (NeEm) across disaster periods.

**Table 1 ijerph-18-05292-t001:** Global univariate Moran’s I for negative emotions across three disaster periods. SD = Standard Deviation, *NeEm*=combined negative emotions. Significance values: *** <0.001, ** <0.01, * <0.05, <0.1.

Disaster Period	Outcome	Moran’s I	z-Value	SD
Pre-disaster	Anger	0.0250 *	2.1205	0.0124
	Disgust	0.0047	0.4543	0.0115
	Fear	0.0210 *	1.7709	0.0120
	Sadness	0.0232 *	1.9617	0.0120
	Shame	−0.0077	−0.5764	0.0131
	NeEm	0.0554 ***	4.7017	0.0120
Peri-disaster	Anger	0.0042	0.3753	0.0124
	Disgust	−0.0027	−0.1795	0.0128
	Fear	0.0125	1.0141	0.0123
	Sadness	−0.0032	−0.2330	0.0122
	Shame	−0.0007	−0.0411	0.0113
	NeEm	−0.0085	−0.6641	0.0124
Post-disaster	Anger	0.0065	0.6001	0.0116
	Disgust	0.0227 *	1.8673	0.0126
	Fear	0.0007	0.1241	0.0113
	Sadness	−0.0050	−0.3967	0.0118
	Shame	0.0202 *	2.1583	0.0097
	NeEm	0.0855 ***	6.9491	0.0123

**Table 2 ijerph-18-05292-t002:** Global bivariate Moran‘s I for exposure variables and combined negative emotions (*NeEm*) across three disaster periods. SD=Standard Deviation, CT= Census Tract. Significance values: *** <0.001, ** <0.01, * <0.05, <0.1.

Disaster Period	Exposure Variable and *NeEm*	Moran’s I	z-Value	SD
Pre-disaster	Unemployment rate	−0.0229 **	−2.5388	0.0088
Peri-disaster	Unemployment rate	−0.0098	−1.0457	0.0093
	>50% flooding in CT	−0.0080	−0.9217	0.0090
	≤50% flooding in CT	0.0014	0.1946	0.0088
	No flooding in CT	0.0051	0.5687	0.0091
	Affected buildings	0.0098	1.0281	0.0091
	Minor building destruction	0.0089	−0.9139	0.0096
	Major building destruction	−0.0065	−0.6927	0.0096
	Destroyed building structures	−0.0051	−0.5452	0.0096
Post-disaster	Unemployment rate	−0.0228 **	−2.5228	0.0090
	>50% flooding in CT	0.0074	0.8067	0.0094
	≤50% flooding in CT	0.0042	0.4985	0.0087
	No flooding in CT	−0.0090	−0.9790	0.0094
	Affected buildings	0.0432 ***	4.7868	0.0091
	Minor building destruction	0.0037	0.3728	0.0095
	Major building destruction	0.0355 **	3.8881	0.0091
	Destroyed building structures	0.0028	0.3411	0.0086

## Data Availability

Research scholars interested in working with the data used in this study are asked to contact the corresponding author with a short description of the planned study. According to our data sharing policies, we then require (1) a data management plan, (2) a signed data sharing and confidentiality agreement, and (3) ethical clearance from an institutional Ethics Committee.

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
