# Peer review of "Space-Time Dependence of Emotions on Twitter after a Natural Disaster"

_ijerph, 2021, doi:10.3390/ijerph18105292_

Round 1
Reviewer 1 Report
This is a well-written manuscript and the results are clear. I have the following comments:
- The authors use the term "natural disaster" in the text, but a natural hazard (such as landslide, earthquake, flood, severe storms, cyclones and so on…) becomes a “disaster” when it has serious impacts on the population both in terms of human and economic losses. I suggest to replace 'natural disaster' with “natural hazard” or “disaster related to natural hazard”.
- In order to avoid confusion for those unfamiliar with the administrative divisions of United States, the five boroughs of NYS must be highlighted in the Figures.
Author Response
This is a well-written manuscript and the results are clear.
Thank you.
I have the following comments: The authors use the term "natural disaster" in the text, but a natural hazard (such as landslide, earthquake, flood, severe storms, cyclones and so on…) becomes a “disaster” when it has serious impacts on the population both in terms of human and economic losses. I suggest to replace 'natural disaster' with “natural hazard” or “disaster related to natural hazard”.
Thank you for this comment. We agree that a natural hazard may become a disaster. This was true for Superstorm Sandy that caused 43 deaths and contributed to $19 billion in damage in NYC alone. Around 63,000 houses were damaged and 300 destroyed. Thousands of residents were left without power, and experienced infrastructural damage (e.g., to public transportation and hospitals) and limited access to necessary resources, including food, water, and healthcare. As this disaster also had mental health consequences for the local population as documented by scholarly research including our own [1,2], we would like to refrain from changing the wording.
In order to avoid confusion for those unfamiliar with the administrative divisions of United States, the five boroughs of NYS must be highlighted in the Figures.
Thank you. We agree and have now adapted the figures.
Reviewer 2 Report
The article is a component of the recently more and more popular trend of studying social media content with Big Data methods. In the variant used by the authors, the techniques of geolocation of tweets are also used.
1. The first question that arises: what made the authors decide to use data from 2012? This data is quite old (we are in 2021).
2. secondly, do the authors see a possibility of using their research method in diagnosing epidemics - as it was done, for example, by Prof. Adam Sadilek from the University of Rochester?
3. the illustrations attached to the text are not very clear. Not much can be deduced from them. Fortunately, more readable versions can be found in the Supplementary files.
4) What has been done to eliminate data distortions (VPN users, proxies, bots and fake accounts)?
5) The conclusions are pretty superficial, overshadowed by Limitations. What I find most missing is a summary of what the cited study shows and what the directions for further research should be.
Author Response
The article is a component of the recently more and more popular trend of studying social media content with Big Data methods. In the variant used by the authors, the techniques of geolocation of tweets are also used.
The first question that arises: what made the authors decide to use data from 2012? This data is quite old (we are in 2021).
This study is part of a series of analyses and adds on previous scientific works [1–7]. Therefore, we have a good understanding about the study population, the local context, and clean datasets on Superstorm “Sandy” including exposure variables from FEMA (Federal Emergency Management Association) available, so that we can also investigate the long-term effects of this natural disaster in affected population in future studies. All this provides the rationale for using the 2012 dataset in this study.
Nevertheless, we have added the following sentences to the limitation section (starting from line 301):
«First, the data are from 2012 and are therefore not recent. Mental health issues, such as PTSD and depression, can linger for long periods of times, even years, if not treated. Therefore, although the data is old in academic terms, the implications can still be relevant to the real live people who have experienced superstorm Sandy. This is especially true for population with low socio-economic status with fewer resources [8]. Twitter as a data source allows for the analysis of the long-term mental health consequences of superstorm sandy in affected users. Furthermore, Twitter is a fairly "prototypical" social platform in a lot of its features and affordances [9], which makes the application of the methodologies used in this study likely transfer well to similar in-the-moment geo-referenced text-based datasets. Future studies could therefore take into consideration changes in Twitter usage since our study to replicate our efforts using more recent data and across more heterogeneous areas/events around the world.»
- secondly, do the authors see a possibility of using their research method in diagnosing epidemics - as it was done, for example, by Prof. Adam Sadilek from the University of Rochester?
This method is very well suited for risk analysis and risk management of disease events. It can be used to identify incipient epidemics and their hotspots and to estimate the spread of epidemics. For an overview in the context of Covid-19 (spatial methods without the use of social media), see e.g., Fatima et. al. [10]. We have also adapted the conclusion section and cited Adam Sadilek’s work [11]. Please also see our response to your question 5.
the illustrations attached to the text are not very clear. Not much can be deduced from them. Fortunately, more readable versions can be found in the Supplementary files.
Thank you. We now provide figures with a better resolution.
4) What has been done to eliminate data distortions (VPN users, proxies, bots and fake accounts)?
Thank you for this important question. We have applied Botometer to identify automated Twitter accounts as described in the methods section. Back in 2012, there were not too many of them as it would be expected in today’s Twitter realm. Furthermore, since we only used geo-referenced tweets, VPN and Proxy users can be ruled out, as VPNs and Proxies are generally used together to mask and hide one’s identity - IP address - in order to not reveal one’s location to a government, or Twitter themself. We have added text in the methodology section to further clarify on this point: p.3 lines 100-104.
5) The conclusions are pretty superficial, overshadowed by Limitations. What I find most missing is a summary of what the cited study shows and what the directions for further research should be.
Thank you very much. We added the following sentences in the conclusion and believe that it is now much clearer (starting from lines 337 and 341):
«We found that low socioeconomic status and infrastructural damage were predominantly correlated with local clusters of negative emotions post-disaster.»
«For example, spatio-temporal triangulation of social media streams with local information on the social and physical contextual situation may provide useful information to estimate the contagion of infectious diseases [11]or mental health consequences of natural and human made disasters [12]. Against the background of increasing use of digitally connected devices and available big social media data, such approaches may help design locally specific mental health relief measures especially during times of crisis and in disaster affected areas.»
Reviewer 3 Report
In general, it seems appropriate to me to measure the consequences of natural disasters on the mental health of the population.
I consider it correct to relate emotional changes to the phases of evolution of a great disaster.
The objectives are relevant and well covered with the social network under study, the methodology and the data analysis. I especially think that the use of modern infographics, with great power to visualize results, adds great value.
The findings seem relevant to act on the population, to avoid reducing the negative impact that any catastrophe, natural or of another nature, can have on the mental health of citizens. The only observation of improvement on my part is how late this work has come, since 2012. And perhaps it is advisable to be cautious since perhaps the results can only be taken into account for future catastrophes in the city of NY or for global cities of the first. world similar to NY.
Author Response
In general, it seems appropriate to me to measure the consequences of natural disasters on the mental health of the population.
Thank you. Indeed, this is appropriate, see for example: [8].
I consider it correct to relate emotional changes to the phases of evolution of a great disaster.
The objectives are relevant and well covered with the social network under study, the methodology and the data analysis. I especially think that the use of modern infographics, with great power to visualize results, adds great value.
The findings seem relevant to act on the population, to avoid reducing the negative impact that any catastrophe, natural or of another nature, can have on the mental health of citizens.
Thank you.
The only observation of improvement on my part is how late this work has come, since 2012.
Thank you. Please see our response to reviewer #2, question 1.
And perhaps it is advisable to be cautious since perhaps the results can only be taken into account for future catastrophes in the city of NY or for global cities of the first. world similar to NY.
We can very well understand your view to be cautious in applying the results to other regions of the world.From our perspective, our approach and the respective findings are applicable to regions of the world where social media is used heavily, since the emotion detecting software and the subsequent spatial approach works also on other textual data that has some geographic information available.
Reviewer 4 Report
I agree that natural disasters can have significant consequences for population mental health.
The study aims were to a) explore the spatial distribution of 17 negative emotional expressions of Twitter users before, during, and after Superstorm Sandy in New 18 York City (NYC) in 2012 and b) examine potential correlations between socioeconomic status and 19 infrastructural damage with negative emotional expressions across NYC census tracts over time. Strengths of the manuscript: good design of research.
Weak parts of the manuscript: only tweets in English language were used.
Research design is presented methodologically correct.
Methods. The tweets were retrieved from the ‘Geo-Tweet’ archive of the ‘Harvard Centre for Geographical Analysis’ (CGA) and missing data were supplemented by data from ‘Geofeedia’. Previously, an emotion analysis of the data was carried out with EMOTIVE (Extracting the Meaning of Terse Information in a Visualization of Emotion), a software that detects and measures emotions in social media posts. This software was used to extract six basic emotions defined by Ekman (anger, dis-95 gust, fear, happiness, sadness and surprise) as well as shame and confusion.
Results. Generally, the number of tweets in the pre- and post-disaster period was approximately the same. In the peri-disaster period however, significantly fewer tweets were posted than in the pre-, and post-disaster periods. The number of tweets that included emotions sadness and fear increased slightly in the peri-disaster period. The emotion sadness was with 46% the most prevalent emotion in terms of absolute numbers of all negative emotions, with about 3%, shame being the least frequently expressed emotion. to predict the choice of emotional style of coping with stress in a group of officers was confirmed.
Limitations: Only tweets in English language were used. Twitter users are not representative of the US population in terms of age, gender, or education, and Twitter users who use the location service are not representative of the general study.
Conclusions: The authors demonstrated how exploratory spatial data analysis can be used to combine geo-referenced Twitter data with publicly available datasets from official sources to identify spatial clusters of negative emotional expressions before, during, and after a natural disaster. It is thus possible to identify areas of increased need for help.
The article written in an appropriate way. The results interpreted appropriately. Article is suitable for publishing.
Author Response
I agree that natural disasters can have significant consequences for population mental health.
The study aims were to a) explore the spatial distribution of 17 negative emotional expressions of Twitter users before, during, and after Superstorm Sandy in New 18 York City (NYC) in 2012 and b) examine potential correlations between socioeconomic status and 19 infrastructural damage with negative emotional expressions across NYC census tracts over time.
Strengths of the manuscript: good design of research.
Thank you.
Weak parts of the manuscript: only tweets in English language were used.
This is correct. However, we think that including also other languages might have introduced additional biases as these would have had to be translated first in order to use the emotion detection software. Therefore, we refrained from also using tweets in other languages as mentioned in the limitation section.
Research design is presented methodologically correct.
Thank you.
Methods. The tweets were retrieved from the ‘Geo-Tweet’ archive of the ‘Harvard Centre for Geographical Analysis’ (CGA) and missing data were supplemented by data from ‘Geofeedia’. Previously, an emotion analysis of the data was carried out with EMOTIVE (Extracting the Meaning of Terse Information in a Visualization of Emotion), a software that detects and measures emotions in social media posts. This software was used to extract six basic emotions defined by Ekman (anger, disgust, fear, happiness, sadness and surprise) as well as shame and confusion.
Results. Generally, the number of tweets in the pre- and post-disaster period was approximately the same. In the peri-disaster period however, significantly fewer tweets were posted than in the pre-, and post-disaster periods. The number of tweets that included emotions sadness and fear increased slightly in the peri-disaster period. The emotion sadness was with 46% the most prevalent emotion in terms of absolute numbers of all negative emotions, with about 3%, shame being the least frequently expressed emotion. to predict the choice of emotional style of coping with stress in a group of officers was confirmed.
Limitations: Only tweets in English language were used. Twitter users are not representative of the US population in terms of age, gender, or education, and Twitter users who use the location service are not representative of the general study.
Conclusions: The authors demonstrated how exploratory spatial data analysis can be used to combine geo-referenced Twitter data with publicly available datasets from official sources to identify spatial clusters of negative emotional expressions before, during, and after a natural disaster. It is thus possible to identify areas of increased need for help.
The article written in an appropriate way. The results interpreted appropriately. Article is suitable for publishing.
Thank you.